# PRESERVE AND PERSONALIZE: PERSONALIZED TEXT-TO-IMAGE DIFFUSION MODELS WITHOUT DISTRIBUTIONAL DRIFT

**Gihoon Kim  Hyungjin Park  Taesup Kim**[†]
Graduate School of Data Science
Seoul National University

## ABSTRACT

Personalizing text-to-image diffusion models involves integrating novel visual concepts from a small set of reference images while retaining the model's original generative capabilities. However, this process often leads to overfitting, where the model ignores the user's prompt and merely replicates the reference images. We attribute this issue to a fundamental misalignment between the true goals of personalization, which are subject fidelity and text alignment, and the training objectives of existing methods that fail to enforce both objectives simultaneously. Specifically, prior approaches often overlook the need to explicitly preserve the pretrained model's output distribution, resulting in distributional drift that undermines diversity and coherence. To resolve these challenges, we introduce a Lipschitz-based regularization objective that constrains parameter updates during personalization, ensuring bounded deviation from the original distribution. This promotes consistency with the pretrained model's behavior while enabling accurate adaptation to new concepts. Furthermore, our method offers a computationally efficient alternative to commonly used, resource-intensive sampling techniques. Through extensive experiments across diverse diffusion model architectures, we demonstrate that our approach achieves superior performance in both quantitative metrics and qualitative evaluations, consistently excelling in visual fidelity and prompt adherence. We further support these findings with comprehensive analyses, including ablation studies and visualizations.

## 1 INTRODUCTION

Recent advances in diffusion-based image generation models have led to remarkable progress in photorealistic and diverse image synthesis (Dhariwal & Nichol, 2021; Ho et al., 2020; Song et al., 2020a; Nichol & Dhariwal, 2021; Ho & Salimans, 2022; Song et al., 2020b; Karras et al., 2022). In particular, text-to-image diffusion models (Rombach et al., 2022; Saharia et al., 2022; Ramesh et al., 2022; Podell et al., 2023; Esser et al., 2024) have demonstrated the ability to synthesize high-quality images across a wide range of prompts, effectively capturing the semantics of general descriptions such as "`a dog in the snow`".

However, these models struggle when prompts involve user-specific or hard-to-describe subject details. In real-world scenarios, users often wish to personalize generation with only a few reference images of their own subjects, which makes the task fundamentally challenging. This motivates the need for personalization methods that can adapt pretrained models to novel, user-specific concepts using only a small number of images (Gal et al., 2022; Ruiz et al., 2023). The ultimate goal of personalization is to faithfully encode new concepts (e.g., "`my dog`") while preserving the model's inherent ability to follow prompts (e.g., "`in the snow`"), thereby generating diverse and compositionally rich images such as "`my dog in the snow`". Unfortunately, learning new subjects from limited data often leads to overfitting where the model disregards prompt content and collapses to simply reproducing the reference images.

---

[†]Corresponding author

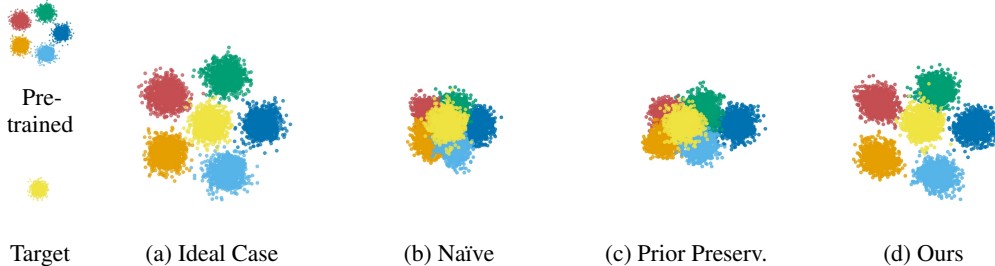

Figure 1: Visualization of training objective differences on the toy dataset for personalization, which are further discussed in Section 4.3. The diffusion model is first trained on the pretrained dataset (five colors) and then personalized with the target dataset (yellow). (a) represents the ideal scenario, where the target data is learned while the original knowledge is preserved. (b) and (c), corresponding to Eqs. 2 and 3, fail to preserve the distributions of all classes. In contrast, (d), which incorporates the proposed regularization, better maintains the structure of the pretrained distribution and best matches the ideal case in (a).

To mitigate this issue, recent methods have attempted to constrain the adaptation process by freezing parts of the network (Hu et al., 2022; Kumari et al., 2023; Tewel et al., 2023) or limiting the magnitude or direction of parameter updates (Qiu et al., 2023; Han et al., 2023). While partially effective, these methods rely on heuristic architectural constraints rather than explicitly considering distributional preservation. As a result, they offer only indirect control over the preservation of the pretrained distribution. Moreover, such techniques often introduce a trade-off between flexibility and stability: freezing too many parameters can prevent proper adaptation to new subjects, while relaxing constraints increases the risk of overfitting and prompt misalignment. These limitations make it difficult to precisely balance the dual objectives of personalization.

In contrast, our approach tackles this challenge at the objective level, providing an explicit mechanism to regularize deviation from the pretrained model without relying on architectural heuristics. By formulating the problem through Lipschitz-based regularization, we offer a principled and continuous control over model updates, enabling better generalization. Figure 1 illustrates the trends arising from the differences between existing and proposed objectives, which are further discussed in Section 4.3. We also show that this formulation can be expressed as a variant of L2 regularization, making it simple to implement yet theoretically grounded. Moreover, our approach does not require a pre-sampling stage for regularization, substantially reducing training time.

We validate our approach through comprehensive experiments across multiple diffusion backbones and baseline personalization methods. Our approach achieves superior performance in both quantitative metrics and qualitative evaluations, excelling in visual fidelity as well as prompt consistency. It also reduces training time by more than half, offering substantial efficiency gains in practice. Furthermore, we present intuitive analyses through visualizations and ablation studies, providing empirical support for our theoretical claims. Our contributions are summarized as follows:

- We identify an objective-level gap in existing personalization approaches and provide a theoretical analysis of its implications.
- We propose a Lipschitz-based regularization objective that guarantees distributional preservation during personalization. The proposed formulation is simple yet theoretically grounded, aligning the training objective with the true goals of personalization.
- Our method demonstrates improvements in both subject fidelity and text alignment, together with clear gains in training time efficiency. We also provide intuitive insights through visual analyses and ablation studies that support the theoretical and empirical findings.

## 2 RELATED WORK

**Personalized Text-to-Image Generation.** A central question of few-shot personalization has been how to adapt pretrained networks to new concepts with only a few subject-specific images. Textual

Inversion (Gal et al., 2022; Voynov et al., 2023) encodes subject-specific information into learned text tokens, while DreamBooth (Ruiz et al., 2023) finetunes the entire network. However, instability in the training process results in mode collapse and overfitting. Subsequent studies aim to improve stability and efficiency by updating subsets of parameters (Meng et al., 2022; Kumari et al., 2023; Tewel et al., 2023) or by applying low-rank adaptation (LoRA) (Hu et al., 2022; Chen et al., 2024). In this context, SVDiff (Han et al., 2023) focuses updates on the singular values of weight matrices, whereas OFT (Qiu et al., 2023) restricts them to angular components. While these enforced constraints help stabilize the training process and preserve generative capacity, the reduced flexibility restricts the model's expressiveness for personalized concepts.

On the other hand, auxiliary conditioners have been employed without modifying the pretrained model. For instance, BLIP-Diffusion (Li et al., 2023a) leverages pretrained vision–language models (Li et al., 2022; 2023b), and IP-Adapter (Chen et al., 2023; Zhang et al., 2023) introduces an adaptor network trained on large-scale datasets. These approaches increase versatility, but they show limited performance on individual personalized concepts. Moreover, they require substantial computational and external resources, making them impractical in few-shot personalization scenarios. Whether through selecting trainable components or designing auxiliary conditioners, these methods remain tied to specific backbone architectures and therefore fall short of providing a general solution.

**Objective Functions for Preserving Distribution.** Objective-level formulations for distribution preservation provide a principled direction for personalization. Prior Preservation Loss (Ruiz et al., 2023) introduces a regularization function via pre-sampling over the generic category of a novel subject, as detailed in Section 3. Subsequent work extends this idea by constraining novel subjects not only at the generic concept level but also at the image level (Qiao et al., 2024) and text level (Huang et al., 2025). However, preserving category-level information does not necessarily imply the preservation of all other prior knowledge, a point we discuss in Section 4.1. Beyond personalization, diffusion-based applications (Poole et al., 2022; Hertz et al., 2023; Wang et al., 2023) introduce objectives to align the output distribution of a learnable generator with that of a pretrained diffusion model. Such methods incorporate pairwise supervision (Yin et al., 2024b) or adversarial objectives (Yin et al., 2024a; Sauer et al., 2024). Their dependence on external datasets and extensive optimization, however, restricts their applicability in personalization settings where only a handful of subject images are available. The limited guarantees of prior methods, together with the poor applicability of alternative approaches to few-shot personalization, point to the need for goal-aligned formulations.

## 3 BACKGROUND

**Text-to-Image Diffusion Models.** Diffusion models are trained to maximize an evidence lower bound (ELBO) on the data log-likelihood $\log p(\mathbf{x})$ due to the intractability (Luo, 2022; Chan et al., 2024; Nakkiran et al., 2024). The generative process is formulated over a sequence of latent variables (Sohl-Dickstein et al., 2015), which can be written as:

$$\log p(\mathbf{x}) = \log \int p(\mathbf{x}_{0:T})\, d\mathbf{x}_{1:T} \;\geq\; \mathbb{E}_{q(\mathbf{x}_{1:T}|\mathbf{x_0})}\left[\log \frac{p(\mathbf{x}_{1:T})}{q(\mathbf{x}_{1:T}\mid \mathbf{x_0})}\right]. \tag{1}$$

Rather than optimizing Eq. 1 directly, DDPM (Ho et al., 2020) adopts a surrogate loss for transition-noise prediction. Text-to-Image diffusion frameworks (Rombach et al., 2022; Saharia et al., 2022; Podell et al., 2023) train the denoiser conditioned on a text prompt $c$, and the resulting conditional surrogate loss maximizes the joint distribution $p_\theta(\mathbf{x}, c)$[1]. In detail, an encoder $\mathcal{E}$ maps an image $\mathbf{x}$ to a latent $\mathbf{z} = \mathcal{E}(\mathbf{x})$, and noisy latents are generated as $\mathbf{z}_t = \sqrt{\bar{\alpha}_t}\,\mathbf{z} + \sqrt{1 - \bar{\alpha}_t}\,\boldsymbol{\epsilon}$, where $\boldsymbol{\epsilon} \sim \mathcal{N}(\mathbf{0}, \mathbf{I})$, $t \sim \mathrm{Uniform}\{1, \ldots, T\}$, and $\bar{\alpha}_t$ denotes the cumulative product of the noise schedule. Training then minimizes the conditional noise-prediction loss:

$$\mathcal{L}_{\text{denoise}} = \mathbb{E}_{\mathbf{z},\, \mathbf{c},\, \boldsymbol{\epsilon},\, t}\big\|\boldsymbol{\epsilon} - \boldsymbol{\epsilon}_\theta(\mathbf{z}_t,\, \mathbf{c},\, t)\big\|_2^2. \tag{2}$$

**Prior Preservation Loss.** The combination of the personalization loss and the prior preservation loss has become a conventional objective for personalization. For the personalization loss $\mathcal{L}_{\text{personalize}}$,

---

[1]This follows from the conditional distribution $\log p_\theta(\mathbf{x} \mid c) = \log p_\theta(\mathbf{x}, c) - \log p(c) = \log \int p_\theta(\mathbf{x}, \mathbf{z}_{1:T}, c)\, d\mathbf{z}_{1:T} - \log p(c)$, where $\log p(c)$ is constant with respect to $\theta$.

the target prompt $c_{\text{target}}$ (e.g., "A photo of V* dog") incorporates the special token V*, which guides the model to learn subject-specific features. For the prior preservation loss $\mathcal{L}_{\text{prior}}$, the class prompt $c_{\text{class}}$ (e.g., "A photo of a dog") is used, and 100~200 class-prompt latents $z'_t$ are sampled from the pretrained model. As both objectives follow the conditional diffusion objective in Eq. 2, the combined formulation is:

$$\mathcal{L}_{\text{total}} = \mathbb{E}_{\mathbf{z}, \mathbf{z}', \mathbf{c}_{\text{target}}, \mathbf{c}_{\text{class}}, \boldsymbol{\epsilon}, t} \left[ \|\boldsymbol{\epsilon} - \boldsymbol{\epsilon}_\theta(\mathbf{z}_t, \mathbf{c}_{\text{target}}, t)\|_2^2 + \lambda_{\text{prior}} \|\boldsymbol{\epsilon} - \boldsymbol{\epsilon}_\theta(\mathbf{z}'_t, \mathbf{c}_{\text{class}}, t)\|_2^2 \right]. \quad (3)$$

## 4 METHOD

In this section, we investigate why the standard objective fails to enable effective learning in personalization. Next, we explore how to provide a distributional bound tailored to preserving the generative capacity of the pretrained distribution. Let $x \in \mathcal{X}$ denote an image and $c \in \mathcal{C}$ its associated text prompt. The pretraining dataset $D_{\text{base}} = \{(x_i, c_i)\}_{i=1}^N$ is sampled i.i.d. from a distribution $p_{\text{base}}(x, c)$ that closely approximates the true data distribution $p^*(x, c)$. The personalization dataset $D_{\text{per}} = \{(x_j, c_j)\}_{j=1}^M$, with $M \ll N$, has underlying distribution $p_{\text{per}}(x, c)$. Such assumptions are standard and align with practical settings, ranging from large-scale pretraining on LAION-400M (Schuhmann et al., 2021; 2022) to personalization datasets with only 4~6 images per subject (Ruiz et al., 2023).

### 4.1 MOTIVATION: OBJECTIVE–GOAL MISALIGNMENT

The goal in personalization is not merely to fit the provided data distribution, but to preserve the generalization capabilities of the pretrained model while incorporating novel subject information. We now demonstrate a fundamental misalignment that standard personalization objectives lead to divergence from the original distribution.

**Theorem 1.** *Let $p^*(x, c)$ denote the reference distribution, and let the model parameters $\theta_{base}$ have distribution $p_{\theta_{base}}(x, c)$ satisfying, for any $\varepsilon > 0$,*

$$\left| p^*(x, c) - p_{\theta_{base}}(x, c) \right| < \varepsilon.$$

*Suppose there exists a measurable set $D \subset \mathcal{X} \times \mathcal{C}$ such that*

$$p_{adapt}(D) = \gamma, \qquad p^*(D) = \delta, \qquad \gamma \gg \delta > 0.$$

*The model is adapted by gradient descent on the denoising loss $\mathcal{L}_{denoise}$ from Eq. 2, trained on $(x, c) \sim p_{adapt}$:*

$$\theta_{t+1} = \theta_t - \eta \nabla_\theta \mathcal{L}_{denoise}(\theta_t), \quad \text{where } \theta_0 = \theta_{base}.$$

*After sufficient iterations $t$, $p_{\theta_t} \longrightarrow p_{adapt}$, under universal approximation and convergence assumptions.*

*Then*

$$D_{KL}(p^* \| p_{\theta_{base}}) < D_{KL}(p^* \| p_{\theta_t}).$$

*Proof.* See Appendix F. □

*Remark* 1. Personalization based on the standard diffusion objective (Eq. 2) provides no guarantee of preserving the pretrained distribution and may fail to preserve the generative capacity, which can potentially lead to overfitting or mode collapse.

**Corollary 1.** *Let $p_{class}$ be the sub-distribution of $p(x, c_{class})$ used for prior preservation as described in Section 3. $p'_{per}$, a mixture of $p_{per}$ and $p_{class}$, still satisfies $M \ll N$. Then,*

$$p'_{per}(D) \geq \gamma', \quad p^*(D) \leq \delta \quad (\gamma' \gg \delta).$$

*By Theorem 1,*

$$D_{KL}(p^* \| p_{\theta_{base}}) < D_{KL}(p^* \| p'_{\theta_t}).$$

*Remark* 2. Even when trained with prior-preservation samples based on the class prompt, as in Eq. 3, the model still offers no guarantee of preserving the pretrained distribution.

**Algorithm 1** Lipschitz-based Regularization for Personalization

**Require:** Pretrained $\epsilon_{\theta_{\text{base}}}$, dataset $\mathcal{D}_{\text{per}}$, prompt $c_{\text{per}}$, weight $\lambda$
**Ensure:** Personalized $\epsilon_{\theta_{\text{per}}}$
~~Prior-preservation pre-sampling:~~
1: ~~for $n = 1, \ldots, N_{\text{cls}}$ do~~
2: ~~$z_{\text{cls}}^{(n)} \sim p_{\epsilon_{\theta_{\text{base}}}}(z \mid c_{\text{cls}}); \mathcal{D}_{\text{cls}} \leftarrow \mathcal{D}_{\text{cls}} \cup \{z_{\text{cls}}^{(n)}\}$~~
3: ~~end for~~
4: $\epsilon_{\theta_{\text{per}}} \leftarrow \text{copyWeights}(\epsilon_{\theta_{\text{base}}}); \text{Freeze } \epsilon_{\theta_{\text{base}}}$
 **Personalization Phase:**
5: **for** step **do**
6:     $x \sim \mathcal{D}_{\text{per}}; z \leftarrow \mathcal{E}(x); c \leftarrow \text{TextEncoder}(c_{\text{per}})$
7:     $t \sim \text{Unif}(\{1, \ldots, T\}), \ \epsilon \sim \mathcal{N}(\mathbf{0}, \mathbf{I})$
8:     $\tilde{z}_t := \sqrt{\bar{\alpha}_t} z + \sqrt{1 - \bar{\alpha}_t} \epsilon$, where $\alpha_t := 1 - \beta_t, \ \bar{\alpha}_t := \prod_{s=1}^{t} \alpha_s$
9:     $\mathcal{L} \leftarrow \|\epsilon - \epsilon_{\theta_{\text{per}}}(\tilde{z}_t, t, c)\|_2^2 + \lambda \cdot \sum_i \|\theta_{\text{per}}^i - \theta_{\text{base}}^i\|_2^2$
10:     Update $\theta_{\text{per}}$ w.r.t. $\nabla \mathcal{L}$
11: **end for**

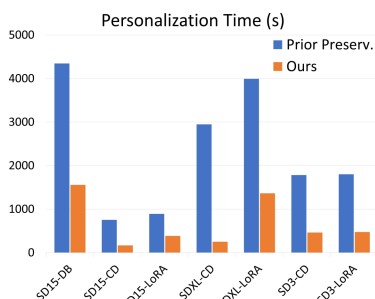

Figure 2: Training efficiency of the proposed method versus the baseline.

A fundamental limitation of existing objectives is revealed by Remarks 1 and 2, highlighting the need for new objectives that can achieve both subject adaptation and distributional preservation. It also explains why the unintended distribution shift poses inherent challenges in personalization. This will be examined as illustrated in Figures 1b and 1c.

## 4.2 LIPSCHITZ-BASED REGULARIZATION FOR DISTRIBUTION PRESERVATION

Motivated by Section 4.1, we investigate a Lipschitz-based regularization that provides an explicit bound on distributional preservation. The following Theorem 2 provides the justification by showing that the generative distribution of the model is bounded under the Lipschitz continuity of the noise prediction network. The detailed proof is deferred to Appendix G.

**Theorem 2.** *If the diffusion model $\varepsilon_\theta$ is Lipschitz continuous in $\theta$, for any two parameter sets $\theta_1$ and $\theta_2$, there exists a constant $\lambda > 0$ such that*

$$D_{KL}\big(p_{\theta_1} \,\|\, p_{\theta_2}\big) \ \leq \ \lambda \cdot \|\theta_1 - \theta_2\|_k. \tag{4}$$

*Proof sketch.*

- Given that the composition of Lipschitz functions preserves Lipschitz continuity (Neyshabur et al., 2015; Asadi et al., 2018) and that attention mechanisms admit finite Lipschitz constants on compact input domains (Castin et al., 2024), we assume $\varepsilon_\theta(x, t)$ is Lipschitz in $\theta$ with finite constant $L_\theta$.

- By Tweedie's formula (Efron, 2011), the score is $s_\theta(x, t) = -\varepsilon_\theta(x, t)/\sigma_t$, and scalar multiplication by $1/\sigma_t$ preserves Lipschitz continuity.

- The probability-flow ODE gives $\log p_\theta(x) = \log p_T(x_T) - \int_0^T s_\theta(x_t, t) \frac{dx_t}{dt} \, dt$ (Song & Ermon, 2019), and since integration is a linear operator, it also preserves the Lipschitz bound.

- Finally, the triangle inequality implies $D_{KL}(p_{\theta_1} \,\|\, p_{\theta_2}) \ \leq \ \lambda \cdot \|\theta_1 - \theta_2\|_k$, where $\lambda = cL_\theta$ for some $c > 0$.

*Remark* 3. For $k = 2$, squaring the inequality yields a form directly connected to an L2 regularization term, $\lambda \cdot \|\theta_1 - \theta_2\|_2 \leq \lambda' \cdot \|\theta_1 - \theta_2\|_2^2$, where $\lambda'$ accounts for the squared scaling.

Building on Theorem 2, we propose a novel objective that regularizes personalization relative to the pretrained distribution based on Eq. 4. Remark 3 demonstrates that minimizing the L2 regularization term serves as a relaxed form of the Lipschitz constraint. Although the L2 form is conventional, its use as a surrogate for Lipschitz continuity in the context of personalization has not been utilized before. The full training procedure with the proposed objective is summarized in Algorithm 1. As a result, the proposed formulation both learns the new subject distribution and preserves the pretrained distribution, which the conventional objective is unable to ensure.

The proposed objective also provides practical advantages. First, preserving the distribution through parameter distance eliminates the need for additional datasets such as prior samples or external resources. As illustrated in Figure 2, removing this requirement yields a considerable reduction

Table 1: Performance gains across backbone models.

| Model | Method | DINO ↑ | CLIP-T ↑ | CLIP-I ↑ |
|---|---|---|---|---|
| SD-1.5 | DB | 0.6028 | 0.2793 | 0.7881 |
| | + Ours | **0.6394 (+0.0366)** | **0.2976 (+0.0183)** | **0.7948 (+0.0067)** |
| | CD | 0.5568 | 0.3154 | 0.7539 |
| | + Ours | **0.5638 (+0.0070)** | **0.3158 (+0.0004)** | **0.7550 (+0.0011)** |
| | DB-LoRA | 0.5776 | **0.3108** | 0.7739 |
| | + Ours | **0.6038 (+0.0262)** | 0.2975 (-0.0133) | **0.7915 (+0.0176)** |
| SD-XL | CD | 0.5337 | 0.3239 | 0.7485 |
| | + Ours | **0.5493 (+0.0156)** | **0.3251 (+0.0012)** | **0.7540 (+0.0055)** |
| | DB-LoRA | 0.6562 | **0.3099** | 0.7970 |
| | + Ours | **0.6819 (+0.0257)** | 0.3014 (-0.0085) | **0.8103 (+0.0133)** |
| SD-3.0 | CD | 0.6168 | **0.3031** | 0.7929 |
| | + Ours | **0.6271 (+0.0103)** | 0.2955 (-0.0076) | **0.8023 (+0.0094)** |
| | DB-LoRA | 0.6043 | 0.3098 | 0.7823 |
| | + Ours | **0.6147 (+0.0104)** | **0.3106 (+0.0008)** | **0.7869 (+0.0046)** |

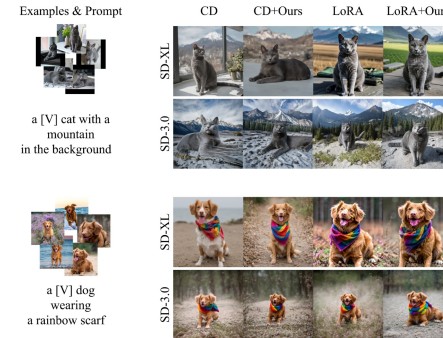

Figure 3: Visual comparison of the proposed method and baselines across backbone models.

in training time. Second, the Lipschitz-based regularizer bounds the KL divergence from the pretrained distribution with a single hyperparameter $\lambda$, thereby quantifying the degree of distributional preservation during training and enabling explicit control as discussed in Section 5.2. By providing a simplified yet explicit mechanism for control, our approach contrasts with previous methods that addressed the issue only implicitly through heuristic strategies. The subsequent experiments validate these advantages and highlight the practical impact of our objective.

### 4.3 Toy Experiment: Method Validation

In this section, we present a visual analysis using toy data to validate the claims established in Theorems 1 and 2. The experiment investigates how the proposed method influences the preservation of the pretrained distribution when the model adapts to a new target, in comparison to existing approaches. We employ diffusion models trained on 2D data sampled from Gaussians with shared variance and different means. As illustrated in Figure 1, the leftmost figures show the pretrained distribution of five classes (top) and an introduced class to be learned (in yellow, bottom). The other four figures present the outcomes of personalization under different objectives.

These results demonstrate that the proposed regularization preserves the pretrained distribution more effectively than conventional objectives. When trained with Eq. 2, the pretrained distribution is not preserved. Using Eq. 3 partially preserves the selected class (shown in deep blue), but the rest of the pretrained classes are not maintained. In contrast, applying our method as in Algorithm 1 enables the model to better preserve the original distribution while still adapting to the new data. This empirical result suggests that, unlike conventional methods which primarily preserve the subject class (e.g., "dog" in "my dog in the snow"), the proposed regularization retains broader contextual generality (e.g., "the snow") during personalization. It further demonstrates that the method can maintain the pretrained distribution without requiring explicit access to the original training data, making it well-suited for personalization scenarios with limited data and complex distributions.

## 5 Experiments and Analysis

We evaluate the proposed method through qualitative and quantitative comparisons with baselines across diverse backbones. We further conduct three ablation studies to examine the practical implications of the proposed formulation in the context of personalized diffusion models. Further analysis, including failure cases, is provided in Appendix L.

**Implementation Details.** We evaluate our method and baselines on the widely used personalization benchmark (Ruiz et al., 2023), which includes 30 subjects, each with up to 6 example images. To assess the generality of the proposed method, we evaluate it across different backbones, ranging from Stable Diffusion v1.5 (SD-1.5) (Rombach et al., 2022) to Stable Diffusion-XL (SD-XL) (Podell et al., 2023) and Stable Diffusion-3.0 (SD-3.0) (Esser et al., 2024). Since both SD-XL and SD-3.0 discourage updating all parameters due to computational inefficiency, we focus our evaluation on parameter-efficient methods. For baseline comparisons, we consider DreamBooth (DB) (Ruiz et al., 2023), Textual Inversion (Gal et al., 2022), Custom Diffusion (CD) (Kumari et al., 2023), SVDiff (Han et al., 2023), and OFT (Qiu et al., 2023), as well as an extension based on LoRA (Hu

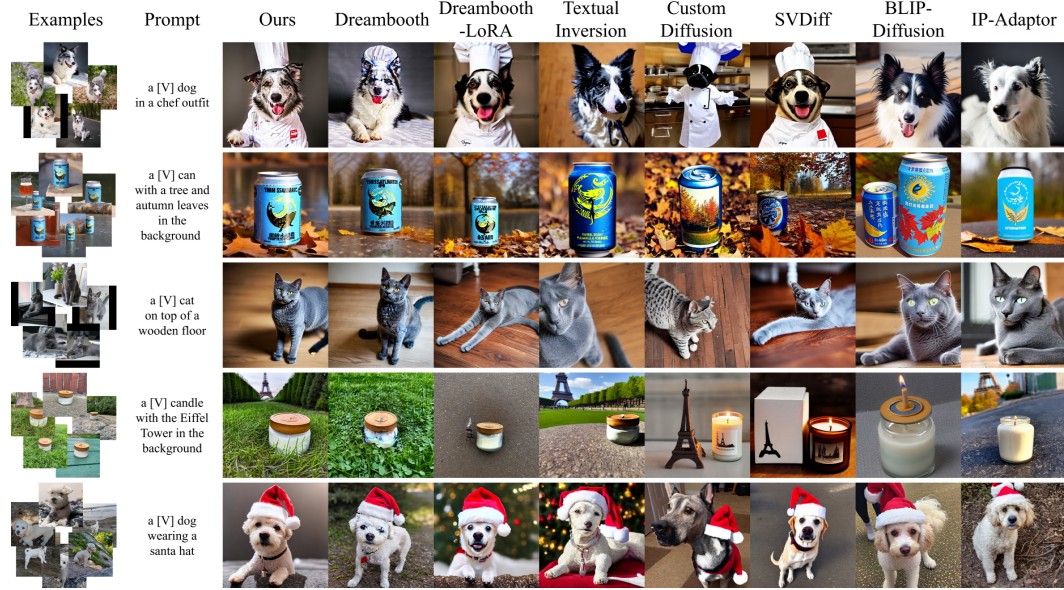

Figure 4: Visual comparison with baseline methods. Ours denotes DreamBooth with the proposed method.

et al., 2022). While we attempted to reproduce OFT using the official codebase, it did not produce usable outputs under our setting. As a result, we report its performance based on the results presented in the original paper. We also include BLIP-Diffusion (Li et al., 2023a) and IP-Adapter (Chen et al., 2023) for comprehensive comparisons. For more details on the setup and implementation of individual experiments, please refer to Appendix H.

**Evaluation Methods.** We evaluate each method based on the images generated from evaluation prompts. For each subject, we use the 25 evaluation prompts provided in the benchmark, generating 4 images per prompt, resulting in 100 images per subject and a total of 3,000 images across all 30 subjects. We evaluate model performance using three metrics: DINO, CLIP-T, and CLIP-I. We use pretrained DINO ViT-S/16 (Caron et al., 2021) and CLIP ViT-B/32 (Radford et al., 2021) models to extract image and text embeddings. CLIP-I and DINO measure subject fidelity by computing the cosine similarity between each generated image and the corresponding input reference images. CLIP-T evaluates text-image alignment based on cosine similarity between the prompt and the generated image. A broader range of results is provided in Appendix L.

## 5.1 EXPERIMENTAL RESULTS

**Comparisons across Backbone Networks.** Table 1 presents quantitative results showing the effect of replacing the baseline objective with the proposed objective across different backbones. The method improves either both image fidelity and text alignment, or improves one without reducing the other. This is remarkable since the two objectives are typically in conflict, where gains in one often lead to losses in the other. In SD-1.5 with DB, the DINO score increases substantially, while the CLIP-T score also improves. Moreover, we observe consistent improvements when the proposed objective is combined with other personalization strategies. For example, CD in SD-1.5 and SD-XL, as well as DB-LoRA in SD-3.0, show gains across all metrics, while the remaining settings still yield meaningful improvements.

A key aspect of the performance improvements is that our method substantially enhances image fidelity, while maintaining text alignment at a high level (around 0.3). This indicates that the proposed method accelerates the learning of novel concepts while preserving the knowledge encoded in the pretrained model. As shown in Figure 3, both baseline and our method produce results that faithfully reflect the input prompts. However, Figure 3 also shows that baseline methods fail to capture the characteristics of the reference image. For example, in the case of CD with SD-XL, the identity of

Table 2: Quantitative comparison of baseline methods.

| Method | DINO ↑ | CLIP-T ↑ | CLIP-I ↑ | Rank |
|---|---|---|---|---|
| Few-shot personalization methods | | | | |
| Ours | **0.6394** | 0.2976 | **0.7948** | **1** |
| DreamBooth | 0.6028 | 0.2793 | 0.7881 | 3 |
| DreamBooth-LoRA | 0.5778 | 0.3095 | 0.7731 | 6 |
| Textual Inversion | 0.5342 | 0.2601 | 0.7539 | 9 |
| Custom Diffusion | 0.5568 | 0.3154 | 0.6479 | 7 |
| SVDiff | 0.3839 | **0.3194** | 0.6886 | 8 |
| OFT | 0.6320 | 0.2370 | 0.7850 | 4 |
| Conditioning with external resources | | | | |
| BLIP-Diffusion | 0.5943 | 0.2865 | 0.7935 | 5 |
| IP-Adaptor | 0.6304 | 0.2635 | 0.8318 | 2 |

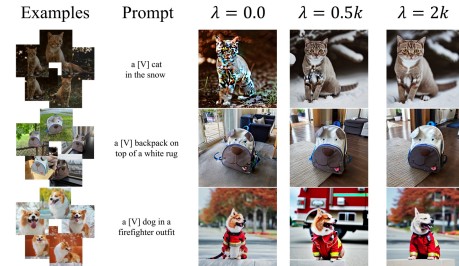

Figure 5: Visual comparison by regularization strength.

the dog is not preserved by the baseline, while our method achieves a clear improvement. These observations are consistent with the quantitative improvements and highlight that the proposed approach contributes both robustness and versatility.

**Comparisons with Baseline Methods.** As shown in Figure 4, we compare the proposed method with various baseline approaches. The proposed method demonstrates superior performance in both fidelity and prompt alignment compared to the baselines. Among the baselines, DB and DB-LoRA show competitive performance. However, DB either fails to incorporate the prompt or exhibits artifacts in the third row. DB-LoRA tends to underrepresent subject identity in the first row. Other methods generally underperform. CD struggles to capture fine-grained subject details. Textual Inversion and SVDiff tend to focus more on prompt alignment than subject fidelity. BLIP-Diffusion and IP-Adapter yield results that preserve the overall subject structure but often miss identity-specific features. In contrast, our method faithfully captures the fine-grained details of the input images while reflecting the text prompts. We conduct the human evaluation in Appendix J, providing a detailed analysis across multiple criteria to assess perceptual aspects as well.

These qualitative observations are consistent with the quantitative results summarized in Table 2. When ranking models jointly on fidelity (DINO and CLIP-I) and alignment (CLIP-T), our method is ranked first. This indicates that the proposed approach provides a more balanced improvement across all metrics compared to the baselines. Since other baselines achieve higher scores on one metric but lower on the other, no baseline simultaneously offers comparable performance on both. For example, while SVDiff achieves the highest CLIP-T score, its overall ranking remains low. These observations reveal a clear trade-off between text-image alignment and subject fidelity. Nonetheless, our method alleviates this trade-off and achieves superior performance in both quantitative and qualitative evaluations.

## 5.2 ABLATION STUDIES

**Balancing Adaptation and Preservation.** We analyze how the proposed regularization affects the quality of personalized outputs. Figure 5 shows the effect of varying the regularization weight $\lambda$. With higher $\lambda$, the generated images align more closely with the prompt but often miss key subject attributes. With lower $\lambda$, they capture subject-specific features, but may collapse to imitating the input images or introduce visual artifacts. A similar pattern is quantitatively confirmed in Figure 6. These experiments investigate the behavior of image fidelity and text alignment under varying regularization strengths. A larger $\lambda$ increases CLIP-T, indicating stronger text alignment and greater preservation of the pretrained distribution. In contrast, a smaller $\lambda$ leads to higher DINO and CLIP-I scores, indicating closer adaptation to the target subject in terms of visual fidelity. This observation suggests that the proposed objective simultaneously addresses both adaptation and preservation while enabling a controllable balance between them.

**Empirical Validation of the Lipschitz Condition.** We assess whether diffusion models follow our claim during personalization under the assumptions. To examine whether the Lipschitz continuity holds in diffusion models, we measure the parameter deviation $\Delta\theta$ and the corresponding output difference $\Delta\epsilon$ between the pretrained and personalized models. Figure 6 illustrates the effect of applying Lipschitz regularization to the personalized diffusion model. As the regularization strength $\lambda$ increases, both $\Delta\theta$ and $\Delta\epsilon$ decrease proportionally. This behavior indicates that the change in the

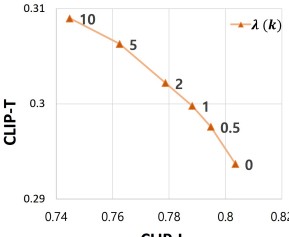 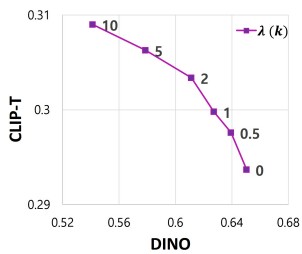 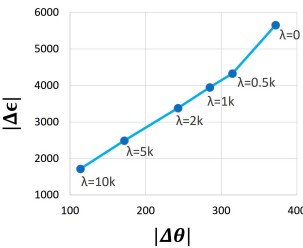

Figure 6: Trade-off curves between image fidelity and text alignment (left and middle) and qualitative output variations (right), both evaluated across different $\lambda$ values.

output of the model is bounded by the change in parameters. The result provides empirical support for our theoretical claim.

**Training Efficiency.** We measure the training time required for personalization to assess the proposed method compared to prior regularization across different backbones. Figure 2 reports averaged results from 10 trials under the same environment for each method. The results shows that our approach is substantially more efficient than methods relying on pre-sampling. The prior preservation strategy consumes the majority of training time for generating prior samples, whereas our algorithm does not require this bottleneck as outlined in Algorithm 1. The improvement is particularly evident in parameter-efficient baselines. Their shorter training time renders the cost of pre-sampling dominant. In SDXL-CD, the training time is reduced to less than one-fifth of the baseline. Other baselines also achieve more than a twofold reduction. As training efficiency is essential for the feasibility of real-world applications (Cho et al., 2024; Chen et al., 2025), such improvements represent a key advantage of the proposed method alongside its performance gains.

## 6 CONCLUSION

In this paper, we demonstrate that existing personalization objectives are misaligned for preventing distributional drift when adapting to novel concepts. Therefore, we introduce a Lipschitz-based regularization objective that bounds distributional shift under a Lipschitz constraint. The proposed formulation is simple yet theoretically grounded, aligning the training objective with the true goals of personalization. Through extensive experiments across diverse backbones and baselines, we validate consistent improvements in both subject fidelity and text alignment. Ablation and visual analyses further highlight the necessity of the regularization, while the method requires no additional time-consuming stages and provides explicit control over the adaptation–preservation trade-off. In conclusion, we contribute a principled approach to personalization that integrates new concepts without unintended deviation from the pretrained distribution. We hope our findings provide a practical reference for balancing adaptation and preservation in post-training across diverse domains.

## LIMITATIONS AND FUTURE WORK

Although our work advances over prior limitations and improves overall performance, it does not guarantee consistent gains across every metric, prompt, or subject. This is because the required degree of adaptation to a novel concept and the level of preservation needed to maintain pretrained behavior can vary across subjects and backbones. Future methods should move beyond retaining prior knowledge and instead focus on better integrating novel subject characteristics to enable more effective and generalizable personalization. These challenges highlight the need for adaptive personalization training strategies (Xuhong et al., 2018; Asadi et al., 2022). One potential direction is to adaptively weight regularization across parameters. In particular, EWC (Kirkpatrick et al., 2017) can assign more precise adaptive weights using the Fisher information $F_i$ for each parameter $\theta_i$. However, $F_i$ is computed (or approximated) from the log-likelihood gradients evaluated on the original training dataset $\mathcal{D}_{\text{pretrain}}$ and the corresponding pretrained parameters $\theta_{\text{pretrain}}$. Because the original pretraining data are not available in practical personalization scenarios, this approach is

difficult to apply in practice. Nevertheless, if Fisher information could be estimated reliably in this setting, it would provide further benefits for adaptive optimization.

## ACKNOWLEDGEMENTS

This work was supported by the National Research Foundation of Korea (NRF) grant funded by the Korea government (MSIT) (No. RS-2025-25424122, Controllable Image Generation in Diffusion Models with Distribution Preservation; No. RS-2024-00345809, Research on AI Robustness Against Distribution Shift in Real-World Scenarios; and No. RS-2023-00222663, Center for Optimizing Hyperscale AI Models and Platforms).

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

APPENDIX

## A ROLE OF $\lambda$ AND PRACTICAL REGULARIZATION IN PERSONALIZATION

Although $\lambda$ is motivated by the Lipschitz property described in Theorem 2, its practical value in personalization is determined by the characteristics of each subject. As shown in line 9 of Algorithm 1, the regularizer coefficient $\lambda$ balances the new learning signal and the preservation of the pretrained capacity. Since learning dynamics differ across subjects, the optimal $\lambda$ varies as well. Moreover, computing the Lipschitz constant of $\theta$ requires access to the pretrained data distribution, whereas personalization provides only a handful of subject images, making it challenging to recover distributional characteristics necessary for estimating the exact Lipschitz constant. This limitation highlights the need for empirical tuning in practice.

We attempt to estimate the Lipschitz constant using a Monte Carlo approximation over $x$ and $t$, combined with a randomized singular-value estimate of the Jacobian for SD1.5. The estimated Lipschitz constant is approximately $184$. As shown in Figure 5, this magnitude tends to prioritize image fidelity over text alignment. Figure 6 further shows that choosing an appropriate value of $\lambda$ is important for balancing these two aspects. This estimated constant is smaller than the reported $\lambda = 500$. We note that this difference arises because, as shown in Algorithm 1, the coefficient $\lambda$ serves not only as a Lipschitz-motivated term but also as a balancing factor between preserving pretrained capacity and learning subject-specific guidance. For this reason, personalization may require using a larger $\lambda$, and doing so does not violate Theorem 2. We also noticed that the theoretical Lipschitz $\lambda$ and the practical objective coefficient $\lambda$ were used with a bit of overlap in terminology, and we will clarify this distinction. We note that choosing a different $\lambda$ only changes the strength of the regularization and does not violate the bound on distribution preservation established in Theorem 2.

Finally, we discuss the norm choice related to Remark 3. We adopt the $\ell_2$ norm as clarified in Remark 3. We observed that using the raw absolute norm $|\cdot|$ leads to unstable updates, whereas using $|\cdot|_2^2$ consistently yields stable training. This observation suggests that our formulation shares a degree of similarity with Kirkpatrick et al. (2017), as the squared $\ell_2$ norm yields larger gradients for larger parameter differences and thus provides a mild form of parameter-wise adaptivity.

## B DISCUSSION

We provide a further discussion on the relationship between evaluation metrics and the claims made in diffusion-based personalization models. Our method introduces a flexible mechanism to balance image fidelity and text alignment. As a result, we observe model variants that score higher in either image fidelity or text alignment. While the main paper is structured around results that show improvements across all metrics, we also observed that some variants with slightly lower text-alignment scores nevertheless produced images that appeared qualitatively satisfactory and well aligned with the prompts. Although this observation is subjective, it raises an interesting dilemma. Demonstrating superior performance over baselines on every metric does not guarantee improved qualitative alignment, and this misalignment highlights a core challenge in personalization. While the proposed approach successfully manages the trade-off between metrics and often produces results that appear qualitatively superior, we emphasized improvements across all quantitative measures to highlight numerical gains. This underscores a broader challenge in diffusion research: the ambiguity and limitations of current metrics. We believe this calls for a deeper examination of how evaluation metrics are used and interpreted in generative modeling.

## C ETHICS STATEMENT

Although our work takes an analytical approach, the broader field of personalized image generation requires careful consideration due to possible misuse. These models could be used to create misleading or harmful images, which may negatively affect individuals or society. This highlights the need to consider ethical implications alongside technical progress in this area of research.

## D    REPRODUCIBILITY STATEMENT

We ensured a controlled and transparent experimental environment to support fair comparison and reproducibility. For baseline comparisons, we followed the original hyperparameter settings and code configurations whenever possible. We fixed random seeds during training and evaluation to further enhance reproducibility. Additional details are provided in Appendix H. We will release the code together with the paper.

## E    LLM USAGE

We employed large language models (LLMs) in a supportive role. We used them primarily for translation, literature search and verification, and for prototyping code consistent with our intentions. The underlying ideas and design were conceived and developed by the authors.

# F    PROOF OF THEOREM 1

**Theorem 1.** Let $p^*(x, c)$ denote the reference distribution, and let the model parameters $\theta_{\text{base}}$ have distribution $p_{\theta_{\text{base}}}(x, c)$ satisfying, for any $\varepsilon > 0$,

$$\left| p^*(x, c) - p_{\theta_{\text{base}}}(x, c) \right| \; < \; \varepsilon.$$

Suppose there exists a measurable set $D \subset \mathcal{X} \times \mathcal{C}$ such that

$$p_{\text{adapt}}(D) = \gamma, \qquad p^*(D) = \delta, \qquad \gamma \gg \delta > 0.$$

The model is adapted by gradient descent on the denoising loss $\mathcal{L}_{\text{denoise}}$ from Eq. 2, trained on $(x, c) \sim p_{\text{adapt}}$:

$$\theta_{t+1} = \theta_t - \eta \nabla_\theta \mathcal{L}_{\text{Denoise}}(\theta_t), \quad \text{where } \theta_0 = \theta_{\text{base}}.$$

After sufficient iterations $t$, $p_{\theta_t} \longrightarrow p_{\text{adapt}}$, under universal approximation and convergence assumptions.

Then

$$D_{KL}\big(p^* \,\|\, p_{\theta_{base}}\big) \; < \; D_{KL}\big(p^* \,\|\, p_{\theta_t}\big).$$

*Proof.* By the three-point property of Bregman divergences Nielsen et al. (2007), we get

$$D_{KL}\big(p^* \,\|\, p_{\theta_t}\big) = D_{KL}\big(p^* \,\|\, p_{\theta_{\text{base}}}\big) + D_{KL}\big(p_{\theta_{\text{base}}} \,\|\, p_{\theta_t}\big) + \big\langle p^* - p_{\theta_{\text{base}}}, \; \log(p_{\theta_t}) - \log(p_{\theta_{\text{base}}})\big\rangle.$$

Applying the Cauchy–Schwarz inequality to the rightmost term,

$$\left| \big\langle p^* - p_{\theta_{\text{base}}}, \log(p_{\theta_t}) - \log(p_{\theta_{\text{base}}})\big\rangle \right| \leq \| p^* - p_{\theta_{\text{base}}} \| \, \| \log(p_{\theta_t}) - \log(p_{\theta_{\text{base}}}) \|.$$

It follows from the convergence assumption that

$$\| \log(p_{\theta_t}) - \log(p_{\theta_{\text{base}}}) \| < M \quad (M < \infty).$$

Then,

$$\| p^* - p_{\theta_{\text{base}}} \| \, \| \log(p_{\theta_t}) - \log(p_{\theta_{\text{base}}}) \| < \varepsilon \, M,$$

the inner-product term is negligible.

Hence,

$$D_{KL}\big(p^* \,\|\, p_{\theta_t}\big) \; \approx \; D_{KL}\big(p^* \,\|\, p_{\theta_{\text{base}}}\big) + D_{KL}\big(p_{\theta_{\text{base}}} \,\|\, p_{\theta_t}\big).$$

Applying Pinsker's inequality Cover & Thomas on the biased region $D$ we obtain

$$D_{KL}\big(p_{\theta_{base}} \,\|\, p_{\theta_t}\big) \; \geq \; 2\,(\gamma - \delta)^2.$$

Then,

$$D_{KL}\big(p^* \,\|\, p_{\theta_{\text{base}}}\big) + D_{KL}\big(p_{\theta_{\text{base}}} \,\|\, p_{\theta_t}\big) \; \geq \; D_{KL}\big(p^* \,\|\, p_{\theta_{base}}\big) + 2\,(\gamma - \delta)^2.$$

Putting everything together, we obtain

$$\begin{aligned}
D_{KL}\big(p^* \,\|\, p_{\theta_t}\big) &= D_{KL}\big(p^* \,\|\, p_{\theta_{\text{base}}}\big) + D_{KL}\big(p_{\theta_{\text{base}}} \,\|\, p_{\theta_t}\big) + \big\langle p^* - p_{\theta_{\text{base}}}, \; p'_{\theta_t} - p'_{\theta_{\text{base}}}\big\rangle. \\
&\approx \; D_{KL}\big(p^* \,\|\, p_{\theta_{\text{base}}}\big) + D_{KL}\big(p_{\theta_{\text{base}}} \,\|\, p_{\theta_t}\big) \\
&\geq \; D_{KL}\big(p^* \,\|\, p_{\theta_{\text{base}}}\big) + 2(\gamma - \delta)^2 \\
&> \; D_{KL}\big(p^* \,\|\, p_{\theta_{\text{base}}}\big).
\end{aligned}$$

Then,

$$D_{KL}\big(p^* \,\|\, p_{\theta_{base}}\big) \; < \; D_{KL}\big(p^* \,\|\, p_{\theta_t}\big),$$

as claimed. $\qquad\qquad\qquad\qquad\qquad\qquad\qquad\qquad\qquad\qquad\qquad\qquad\qquad\qquad\qquad\quad$ $\square$

## G PROOF OF THEOREM 2

**Theorem 2.** If the diffusion model $\varepsilon_\theta$ is Lipschitz continuous in $\theta$, then for any two parameter sets $\theta_1$ and $\theta_2$, there exists a constant $\lambda > 0$ such that

$$D_{KL}\big(p_{\theta_1} \,\|\, p_{\theta_2}\big) \;\leq\; \lambda \cdot \|\theta_1 - \theta_2\|_k. \tag{5}$$

*Proof.* Given that the composition of Lipschitz functions preserves Lipschitz continuity (Neyshabur et al., 2015; Asadi et al., 2018) and that attention mechanisms admit finite Lipschitz constants on compact input domains (Castin et al., 2024), we assume $\varepsilon_\theta(x, t)$ is Lipschitz in $\theta$ with finite constant $L$. Hence for all $x, t$ and any $\theta_1, \theta_2$,

$$\big\|\varepsilon_{\theta_1}(x, t) - \varepsilon_{\theta_2}(x, t)\big\| \;\leq\; L \,\|\theta_1 - \theta_2\|_k.$$

By Tweedie's formula Efron (2011); Ho et al. (2020); Song et al. (2020b),

$$s_\theta(x, t) = -\frac{\varepsilon_\theta(x, t)}{\sigma_t}.$$

Scalar multiplication preserves Lipschitz continuity, so

$$\|s_{\theta_1}(x, t) - s_{\theta_2}(x, t)\| \leq L' \,\|\theta_1 - \theta_2\|_k.$$

It follows that $s_\theta$ is Lipschitz in $\theta$ with constant $L' = L \sup_t \sigma_t^{-1}$.[2]

By the probability-flow ODE formulation[3] Song et al. (2020b); Song & Ermon (2019),

$$\log p_\theta(x) = \log p_T(x_T) - \int_0^T s_\theta(x_t, t)\, \frac{dx_t}{dt}\, dt.$$

Since $\log p_T(x_T)$ is constant with respect to $\theta$ and integration preserves Lipschitz continuity,

$$\big|\log p_{\theta_1}(x) - \log p_{\theta_2}(x)\big| \;\leq\; L'' \,\|\theta_1 - \theta_2\|_k \quad for\ \forall x \in X$$

holds with $L'' = \int_0^T L' dt = L \int_0^T \sigma_t^{-1}\, dt$.

Applying the triangle inequality to the definition of KL divergence then gives

$$D_{KL}(p_{\theta_1} \| p_{\theta_2}) = \int p_{\theta_1}(x) \big(\log p_{\theta_1}(x) - \log p_{\theta_2}(x)\big)\, dx$$

$$\leq \int p_{\theta_1}(x) \big|\log p_{\theta_1}(x) - \log p_{\theta_2}(x)\big|\, dx$$

$$\leq L'' \,\|\theta_1 - \theta_2\|_k.$$

This completes the proof with $\lambda = L''$. □

---

[2]In practice, $\sigma_t$ is never exactly zero: implementations either avoid the endpoint(s) or enforce a small floor $\sigma_t \geq \sigma_{\min} > 0$ for numerical stability. Hence $\sup_t \sigma_t^{-1} < \infty$ in our setting.

[3]Following standard practice, we omit the prompt $c$ and write $\varepsilon_\theta(x, t)$ and $\log p_\theta(x)$. All statements extend directly to the conditional case by replacing $\varepsilon_\theta(x, t)$ with $\varepsilon_\theta(x, c, t)$ and $\log p_\theta(x)$ with $\log p_\theta(x, c)$.

## H  ADDITIONAL IMPLEMENTATION DETAILS

**Ours.** All experiments are conducted on a single NVIDIA RTX 3090 GPU. We use the special token `sks`, or `new1` depending on the baseline. For the proposed Lipschitz-bound regularization, we compare both the $L_1$ and $L_2$ norm variants and observe no meaningful difference in performance; accordingly, we adopt the more widely used $L_2$ norm. We also experiment with integrating Dream-Booth's prior-preservation loss but do not obtain improvements, suggesting a conflict between the objectives.

In experiments across different methods, we primarily used DreamBooth on SD 1.5, as this backbone is the most widely adopted and thus provides a fair basis for comparing diverse approaches. The corresponding hyperparameters are listed above, and other methods were also evaluated on the same backbone for consistency. For DreamBooth, we re-implemented the official codebase to allow more detailed experimentation. This ensured fixed random seeds and controlled variables for fair comparison. Finally, to measure the time required for the personalization process, we performed ten runs for each backbone–method combination and report the average.

In experiments across different backbones, we aimed to evaluate the effect of the proposed objective under comparable settings using existing codebases. The hyperparameters were as follows. For DreamBooth on SD-1.5, we set the regularization weight to $\lambda = 500$, trained with a batch size of 1 and a learning rate of $2 \times 10^{-6}$ for 1,000 iterations. For Custom Diffusion on SD-1.5, we used $\lambda = 175$ with a learning rate of $3 \times 10^{-5}$ for 250 iterations. For SD-XL, we considered two settings: $\lambda = 50$ with a learning rate of $5 \times 10^{-5}$ for 250 iterations, and $\lambda = 10$ with a learning rate of $8 \times 10^{-5}$. For SD-3.0, we used a learning rate of $1 \times 10^{-4}$, $\lambda = 10.1$, and 500 iterations. For LoRA, the settings were as follows: on SD-1.5, we used a learning rate of $1 \times 10^{-4}$, $\lambda = 1 \times 10^{-7}$, and 500 iterations; on SD-XL, a learning rate of $5 \times 10^{-4}$, $\lambda = 1 \times 10^{-6}$, and 500 iterations; and on SD-3.0, a learning rate of $4 \times 10^{-4}$, $\lambda = 5 \times 10^{-6}$, and 700 iterations. The choice of backbone and personalization method leads to different hyperparameter settings and training behaviors. The code is available at https://github.com/rlgnswk/Preserve-and-Personalize.

**Baselines.** We provide additional implementation details for the baselines used in our experiments in Section 5. The proposed method and DreamBooth are implemented and trained based on the Diffusers library[4]. For Textual Inversion Gal et al. (2022)[5], Custom Diffusion Kumari et al. (2023)[6], SVDiff Han et al. (2023)[7], and LoRA Hu et al. (2022)[8], we use available implementations and follow their default training settings. OFT Qiu et al. (2023)[9] is trained using the official codebase, but we observed poor performance and were unable to fully reproduce the reported results; hence, we cite the numbers from the original paper. BLIP-Diffusion Li et al. (2023a)[10] and IP-Adapter Chen et al. (2023)[11] are evaluated using the released model weights on our benchmark through direct inference.

Moreover, experiments with different backbone models are conducted using the Diffusers library. For Stable Diffusion-XL (Podell et al., 2023), we follow the same Custom Diffusion reference as above. Since no official implementation of Custom Diffusion for Stable Diffusion-3.0 (Esser et al., 2024) is available, we reimplemented this variant ourselves. For the LoRA version of Stable Diffusion-3.0, we followed the official implementation[12].

**Ablation Studies.** We provide additional implementation details for the ablation studies on our proposed personalization process in Section 5.2. Following the same training configuration as in Section 5, we fix the learning rate and number of training iterations, varying only the regularization strength $\lambda$ for the experiments measuring changes from the pretrained model (i.e., $\Delta\theta$ and $\Delta\epsilon$). For

---

[4] `https://github.com/huggingface/diffusers`

[5] `https://github.com/huggingface/diffusers/tree/main/examples/textual_`
`inversion`

[6] `https://github.com/huggingface/diffusers/tree/main/examples/custom_`
`diffusion`

[7] `https://github.com/mkshing/svdiff-pytorch`

[8] `https://github.com/huggingface/peft/tree/main/examples/lora_dreambooth`

[9] `https://github.com/zqiu24/oft`

[10] `https://github.com/salesforce/LAVIS/tree/main/projects/blip-diffusion`

[11] `https://github.com/tencent-ailab/IP-Adapter`

[12] `https://github.com/huggingface/diffusers/blob/main/examples/`
`dreambooth/train_dreambooth_lora_sd3.py`

$\Delta\epsilon$, we measure the L2 distance between outputs generated using the unconditional class token from the pretrained and personalized models. The reported value is the accumulated L2 distance over all sampled outputs.

**Toy Experiment.** We validate our method using the toy 2D conditional diffusion model as described in Section 4.3. The setup generates 1,000 samples per class from 2D Gaussian distributions with a standard deviation of 0.5. Class labels range from 0 to 4, positioned at the vertices of a regular pentagon. A new sixth class (class 5) supports finetuning. The noise schedule follows a cosine-based formulation with $T = 100$ timesteps. The model uses a one-hidden-layer MLP with ReLU activation and class/time embeddings. Initial training runs for 1,000 iterations to capture the base distribution effectively. Training for the new class is conducted for 5,000 iterations to ensure convergence within the experimental setup. We adopt a DreamBooth-like setup by jointly training the new class with class 0 (deep blue) data. Considering the problem's complexity, we assign a large weight 100 to class 0 in order to clearly observe the effect of different objectives. When using smaller weights, the other classes tend to collapse toward the new data distribution. However, with weight 100, the objective's influence on class 0 becomes apparent. For our proposed method, we apply Lipschitz regularization with respect to the pretrained weights, using a regularization strength of $\lambda = 50$. Even with a relatively small weight (e.g., 1) on the regularization term, its effect is clearly observable in this setup as shown in Section I.

# I   ABLATION ON THE TOY EXPERIMENT

We conduct an ablation on the toy experiment introduced in Section 4.3 to visualize the trend of the proposed method. As shown in Figure 7, the relationship between the newly learned distribution and the original one varies with the regularization strength $\lambda$. Without Lipschitz regularization, all classes tend to collapse toward the newly introduced distribution. Notably, even with $\lambda = 1$, the regularization effect becomes evident. In addition, increasing $\lambda$ progressively enforces preservation of the original distribution, and the newly learned class is mapped closer to one of the existing modes. These results demonstrate that the proposed objective effectively encourages distribution preservation and supports the balancing behavior observed in Section 5.2. Figure 8 provides additional visual comparisons across different values of $\lambda$. Similar to the earlier analysis, these results highlight how the regularization interpolates between adapting to new concepts and preserving prior distributions.

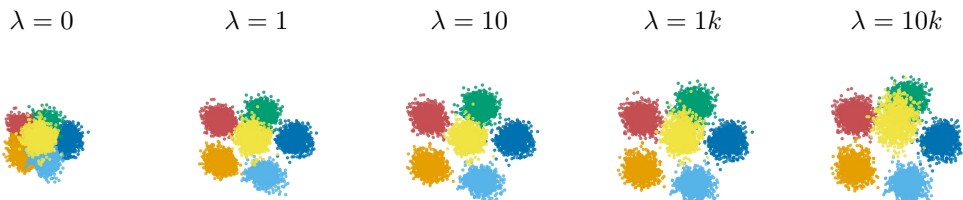

Figure 7: Visualization results under different regularization strengths $\lambda$.

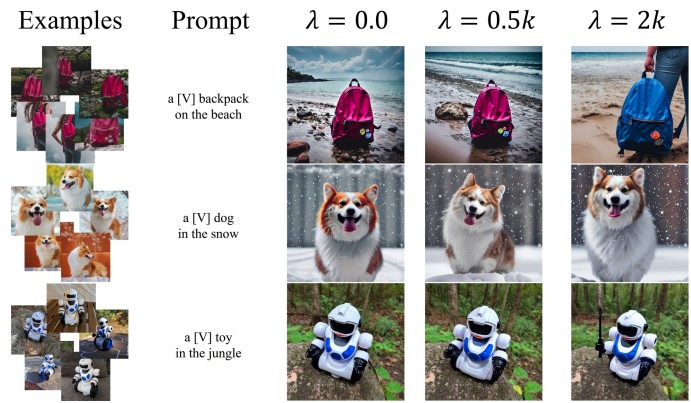

Figure 8: Additional visual comparison by regularization strength.

## J   USER PREFERENCE STUDY

We conducted a user preference study to further support the proposed method.

### J.1   SETUP

A total of 35 participants were asked to select the most preferred image among the model outputs generated from the given test prompts: 35 for general preference and 15 for the detailed criteria evaluation. We designed 30 questions covering all classes in the dataset Ruiz et al. (2023), and randomized the order of the models for each class when presenting them to the participants. We compared four models: our method, DreamBooth (DB) Ruiz et al. (2023), DreamBooth LoRA (DB LoRA) Hu et al. (2022), and the conditioner-based IP-Adaptor Chen et al. (2023), chosen based on their competitive quantitative performance.

The evaluation criteria were explained to the participants in advance, and they were instructed to consider the following aspects comprehensively:

- Identity Preservation: whether the identity of the user-specified object is well preserved.
- Prompt Coherence: whether the condition described in the prompt is accurately reflected.
- Image Quality: whether the image is free of artifacts or distortions.
- Overall Preference: jointly considers the three criteria above.

### J.2   EXPERIMENTAL RESULTS

The results of the user preference study are shown in Table 3. Our method achieved the highest overall preference at 45.81%, while IP-Adaptor recorded the lowest preference at 11.53%. This is in contrast to their positions on the DINO score, where both models ranked among the highest in the quantitative experiment of our submission. We attribute this discrepancy to the IP-Adaptor's inability to accurately represent the target identity, as shown in Figure 4 of our manuscript. DB and DB-LoRA received preference scores of 25.14% and 17.52%, respectively.

The detailed criteria enable a more fine-grained interpretation of the overall preference and reveal clear tendencies of each method. DB shows strong preference in identity preservation but lower prompt adherence. In contrast, DB-LoRA shows lower identity preservation but twice higher prompt adherence. As discussed in Section 5, these tendencies are consistent with the known behaviors of the two methods: DB tends to overfit, whereas DB-LoRA tends to underfit. IP-Adaptor produces good image quality but remains weak in personalization. Our method maintains a balanced performance across all criteria. This suggests that the proposed objective captures subject-specific information while preserving the pretrained distribution, leading to consistent and stable results.

Table 3: User study results across four criteria (all values in %).

| Method | Identity Preservation | Prompt Coherence | Image Quality | Overall Preference |
|---|---|---|---|---|
| Ours | 44.22 | 43.78 | 44.44 | 45.81 |
| DB | 33.33 | 23.11 | 24.89 | 25.14 |
| DB-LoRA | 11.78 | 22.67 | 13.11 | 17.52 |
| IP-Adaptor | 10.67 | 10.44 | 17.56 | 11.53 |

# K  ADDITIONAL COMPARISON WITH DREAMBOOTH

We conduct an additional experiment using DreamBooth. It is the most comparable method since it fine-tunes all parameters as our method does. To compare the effect of the objective, we unify all hyperparameters except the loss. We set the learning rate of DreamBooth to $2 \times 10^{-6}$, the same as ours, instead of its default $5 \times 10^{-6}$. The results are shown in Table 4. Lowering the learning rate in DreamBooth leads to an imbalanced outcome. It improves the CLIP-T score, but lowers both DINO and CLIP-I scores. This suggests reduced image fidelity. As shown in Figure 9, the visual results fail to preserve subject identity in Rows 1 and 3, although this is not always the case, as seen in Row 2. These findings highlight the difference in objectives. Our method achieves better trade-offs and is more suitable for personalization.

Table 4: Additional quantitative comparison of personalization methods. All scores are identical to those reported above, except for the last row.

| Method | DINO ↑ | CLIP-T ↑ | CLIP-I ↑ |
|---|---|---|---|
| Ours | 0.6394 | 0.2976 | 0.7948 |
| DreamBooth | 0.6028 | 0.2793 | 0.7881 |
| DreamBooth (LR $2 \times 10^{-6}$) | 0.5707 | 0.3104 | 0.7695 |

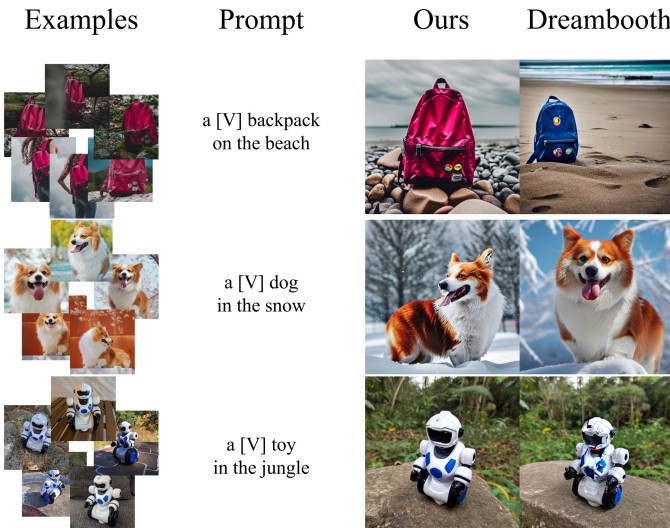

Figure 9: Additional visual comparison with other baselines.

## L ADDITIONAL VISUAL COMPARISON AND FAILURE CASES

We present additional qualitative results and failure cases. As shown in Figure 10, our method captures both subject-specific features and prompt semantics. In contrast, DreamBooth often produces artifacts or fails to reflect the prompt, while DB-LoRA tends to preserve the prompt but underrepresents subject identity. Other baseline methods generally struggle to produce coherent results. We also highlight representative failure cases. Figure 11 shows a case that struggles with cartoon-style minor concepts. The model learns the subject's color but fails to reconstruct its structural identity similar to other methods. In Figure 12, the model fails to properly apply the prompt to the subject. This further shows that prompts requiring changes in the subject's appearance are difficult to handle. These cases indicate that our method also faces challenges in generating certain subject–prompt combinations. As discussed in Section 6, this underscores the importance of incorporating the distributional structure of both the pretrained model and the target subject.

We also present additional results on SD-XL and SD-3.0 in Figure 13. Both backbone models are advanced architectures, and the overall image quality is generally high. For SD-XL, Custom Diffusion fails to preserve subject identity, which is reflected in the low image fidelity scores. While CD+Ours shows improvements, certain cases (e.g., the first row) remain insufficient. Similarly, LoRA exhibits identity shifts in the first row, whereas LoRA+Ours successfully preserves the subject identity, consistent with the improved image fidelity scores. Although CLIP-T remains stable even when it drops in Table 1, some failure cases can still be observed. For example, in Figure 13, the second row of SD-3.0 with CD+Ours does not reflect the prompt, and the third row shows only partial adherence. A similar pattern appears in SD-XL LoRA+Ours, where the prompt is partially captured in the third row. These examples illustrate situations where the text alignment weakens during personalization, which aligns with the observed drop in CLIP-T. We note that although our method improves most metrics and visual results, careful handling is still required across different models and subjects.

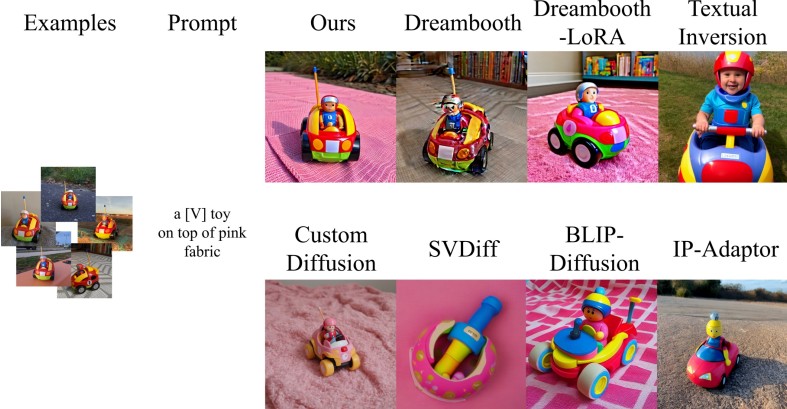

Figure 10: Additional visual Comparison with other baselines.

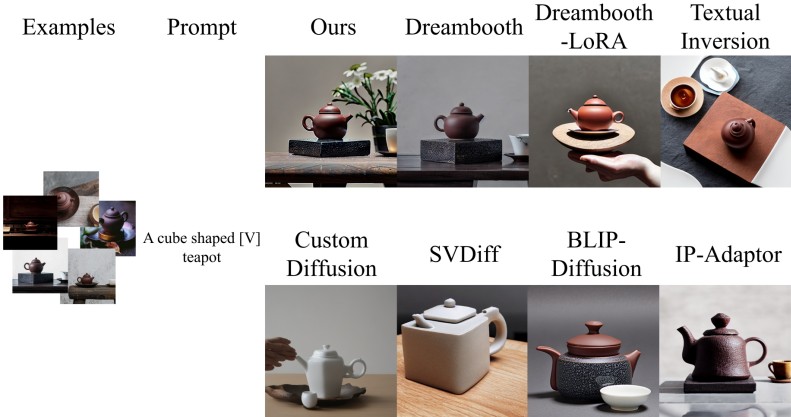

Figure 11: Visual comparison of failure cases across baselines.

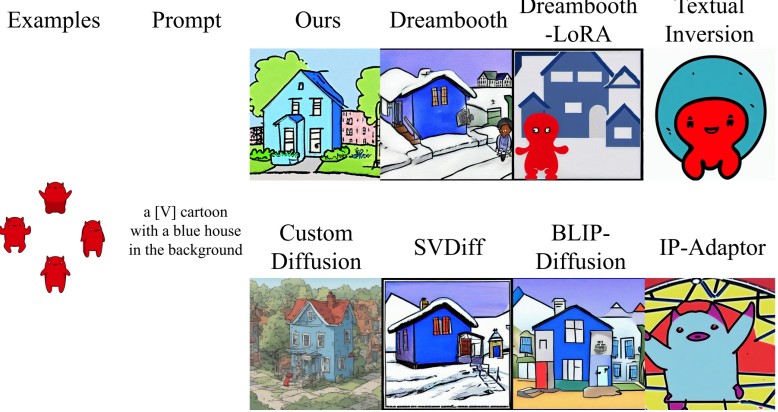

Figure 12: Visual comparison of failure cases across baselines 2.

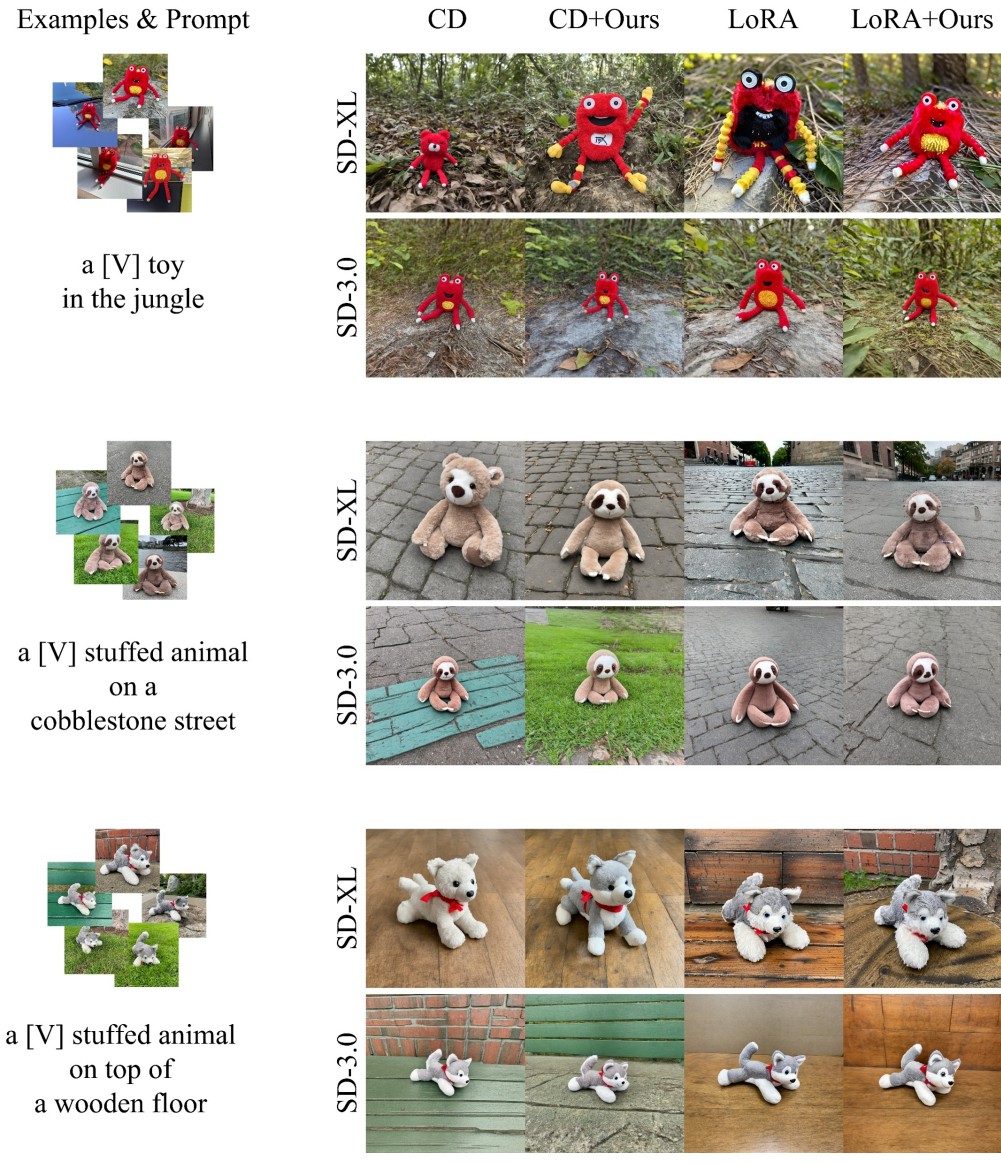

Figure 13: Visual comparison of failure cases across different backbones.

