# OpenReview forum: "Preserve and Personalize: Personalized Text-to-Image Diffusion Models without Distributional Drift"
_ICLR.cc/2026/Conference — ICLR 2026 Poster_

### Official Review · Reviewer_Bv3p · 2025-10-21

**Soundness:** 2
**Presentation:** 2
**Contribution:** 2
**Rating:** 6
**Confidence:** 4

**Summary:**

The paper tackles distributional drift in few-shot personalization of text-to-image diffusion models by adding a Lipschitz-motivated parameter-distance regularizer to the personalization loss. The authors argue standard objectives (and even prior-preservation) don’t explicitly preserve the pretrained distribution (Theorem 1), and show that if the denoiser is Lipschitz in parameters, bounding $|\theta - \theta_{base}|$ controls KL shift (Theorem 2).

**Strengths:**

1. Clear statement of objective–goal misalignment; neat, didactic Theorem 1 showing why standard diffusion fine-tuning drifts.
2. Theory sketch linking parameter Lipschitzness → KL control (Theorem 2); connects elegantly to L2 surrogate.
3. Consistent empirical lift on fidelity (DINO/CLIP-I) with competitive CLIP-T across multiple backbones

**Weaknesses:**

1. The objective is effectively weight decay to $θ_{base}$; relation to continual-learning regularizers (e.g., EWC) and why your variant is preferable in diffusion personalization should be made explicit.
2. You note CLIP-T may drop when fidelity improves; more analysis on prompt adherence failures (per failure figs) would help.
3. In Table 1, SD-3.0 / “+Ours” row shows “CLIP-I +0.0022” but one entry reads “0.0094”. Why is this error proportion so large?

**Questions:**

Can you quantify Theorem 2’s $\lambda$ (constant) for a given backbone (e.g., SD-1.5) via empirical Lipschitz estimation, and relate it to your best-performing $\lambda$ in training?

---

> ### Author Response · Authors · 2025-11-19
> **Response to Reviewer Bv3p**
>
> **Q1. The Proposed Method and Future Direction**
>
> As the reviewer notes, EWC can assign more precise adaptive weights using the Fisher information $F_i$ for each parameter $\theta_i$. However, $F_i$ is computed (or approximated) from the log-likelihood gradients evaluated on the original training dataset $D_{\text{pretrain}}$ and the corresponding pretrained parameters $\theta_{\text{pretrain}}$. Because the original pretraining data are not available in practical personalization scenarios, this approach is difficult to apply in practice. Nevertheless, if Fisher information could be estimated reliably in this setting, it would provide further benefits for adaptive optimization.
>
> We share the reviewer's view that an adaptive mechanism is desirable, and our formulation has a degree of similarity to EWC. As clarified in Remark 3, we derive our regularizer using the squared $\ell_2$ norm, which yields larger gradients for larger parameter differences and thus provides a mild form of parameter-wise adaptivity. Empirically, we observed that training becomes more stable with $|\cdot|_2^2$ than with $|\cdot|$, which supports this behavior.
>
> As noted in Appendix A (Limitations), we consider advanced adaptive mechanisms to be beneficial. For this reason, we also discuss approaches related to EWC. Advancing adaptive strategies tailored to personalization represents an important direction for future research.
>
>
> &nbsp;
>
> **Q2. Analysis of Failure Cases**
>
> We agree that the analysis of failure cases should be clarified. The figures are included in Appendix L, but the main text does not clearly reference each figure. We will make these connections more explicit. Although CLIP-T remains stable even when it drops (line 352), some failure cases can still be observed. For example, in Figure 13, the second row of SD3.0 with CD+Ours does not reflect the prompt, and the third row shows only partial adherence. A similar pattern appears in SD-XL LoRA+Ours, where the prompt is only partially captured in the third row. These examples illustrate situations where the text alignment weakens during personalization, which aligns with the observed drop in CLIP-T.
>
> &nbsp;
>
> **Q3. Typo in Table 1**
>
> Thank you for pointing out the typo. The correct value for our method should be 0.8023, which is an improvement over the previous result of 0.7929. We will correct this in the revised version.
>
> &nbsp;
>
> **Q4. Empirical Lipschitz Estimation**
>
> We estimate the Lipschitz constant in Theorem 2 using a Monte Carlo approximation over $x$ and $t$, combined with a randomized singular-value estimate of the Jacobian for SD1.5. The estimated Lipschitz constant is approximately $184$. As shown in Figure 6, this magnitude tends to prioritize image fidelity over text alignment. Figure 5 further shows that choosing an appropriate value of $\lambda$ is important for balancing these two aspects.
>
> This estimated constant is smaller than the reported $\lambda = 500$. We note that this difference arises because, as shown in Algorithm 1, the coefficient $\lambda$ serves not only as a Lipschitz-motivated term but also as a balancing factor between preserving pretrained capacity and learning subject-specific guidance. For this reason, personalization may require using a larger $\lambda$, and doing so does not violate Theorem 2. We also noticed that the theoretical Lipschitz $\lambda$ and the practical objective coefficient $\lambda$ were used with a bit of overlap in terminology, and we will clarify this distinction.
>
> Another related point is that the Lipschitz constant with respect to $\theta$ depends on both $x$ and $t$, which makes it difficult to compute exactly in a personalization setting. As mentioned in Q1, this limitation arises because estimating the Lipschitz constant with respect to $\theta$ fundamentally requires access to the original training data $(x, c)$, which is unavailable in personalization.
>
> &nbsp;
>
> **Closing Remark**
>
> The reviewer's comments helped clarify the scope of our work and the directions we should further explore. We will address these points carefully in the final manuscript, and we appreciate the reviewer's thoughtful feedback.

---

### Official Review · Reviewer_xLt7 · 2025-10-31

**Soundness:** 3
**Presentation:** 2
**Contribution:** 2
**Rating:** 4
**Confidence:** 4

**Summary:**

This paper focuses on the personalization task in the field of image generation. The authors proposed a method to introduce a Lipschitz-based regularization objective that constrains parameter updates during personalization. The experiment shows the method can achieve competitive performance for this task.

**Strengths:**

- The presentation of figures is great and easy to understand.
- The math notations in this paper are self-contained and well-defined.
- The paper writing is easy to follow.
- The introduction section is good and clearly explains the motivation and the main claims of the paper.
- The proposed method is straightforward.

**Weaknesses:**

- I have to say that tuning-based personalization (like, DreamBooth, Custom Diffusion) is outdated. Learning-based personalization is the mainstream in the current image generation community.
- How did you obtain the results of Fig. 2, or is it just an illustration?
- Could you explain more about the claim "Remark 1. Personalization based on the standard diffusion objective (Eq. 2) provides no guarantee of preserving the pretrained distribution and may lead to divergence"? It is a little bit confusing.
- The evaluation is all based on UNet-based diffusion models. I am curious about the comparison results based on the DiT-based model.
- The evaluation lacks the image quality metrics that are also important.
- There are also some previous works exploring the optimization of personalization learning processes. Have you considered comparing the proposed methods with them?

**Questions:**

Please see the section of weaknesses.

---

> ### Author Response · Authors · 2025-11-19
> **Response to Reviewer xLt7**
>
> **Summary of Our Contribution.** Our work contributes a theoretical analysis of the personalization objective, providing an understanding of how tuning-based optimization operates under diffusion dynamics. Beyond empirical improvements, the proposed objective is principled, backbone-agnostic, and broadly applicable. This perspective offers a broader understanding of how diffusion models can be effectively used and adapted.
>
> &nbsp;
>
> **Q1. Tuning-Based Personalization and Learning-Based Personalization**
>
> We interpret the reviewer's use of "learning-based" as approaches that train a new conditioner or additional network components. Our response focuses on (1) the difference in setting and (2) the remaining need for tuning-based personalization.
>
> - (1) Difference in Setting
>
> Our method assumes a few-shot personalization scenario, where only 4–6 subject images and a pretrained diffusion model are available. In contrast, learning-based approaches generally require larger datasets, more compute, or additional modules such as LLMs or MLLMs [1, 2] to train new conditioning mechanisms. While we acknowledge that learning-based approaches have clear advantages, they are often tied to a specific network design, which limits their generality (lines 110–112).
>
> - (2) Need for a Tuning-Based Method
>
> We agree that successfully trained learning-based approaches can offer strong performance. However, they still tend to lose high-frequency details [3, 4], and they do not provide a way to incorporate information beyond the distribution they were trained on. For this reason, we believe that tuning-based personalization is needed both overall and to a more specific extent, as it can adapt subject-specific information directly from the provided examples. We also consider hybrid strategies that combine learning-based and tuning-based methods (i.e., optimization-based meta-learning) to be a promising direction for future research [5].
>
> &nbsp;
>
> **Q2. Experimental Setup of Fig. 2**
>
> Fig. 2 is designed to compare the time required for each method. We measured the runtime 10 times under the same environment and reported the average. We will make this setting explicit in the revised version. We appreciate the reviewer's detailed feedback.
>
> &nbsp;
>
> **Q3. Explanation of Remark 1**
>
> Remark 1 is intended to explain that existing personalization methods fail to preserve the original generative capacity of the pretrained model. However, this does not imply that the model will necessarily diverge beyond a certain level. For this reason, we used a more cautious wording in the original text. This behavior is also supported by our experiments, where the text alignment is often not maintained after personalization. We agree that the original phrasing can be unclear, and we will revise "may lead to divergence" to a more precise description such as "may fail to preserve the generative capacity, which can potentially lead to overfitting or mode collapse."
>
> &nbsp;
>
> **Q4. Results on DiT-Based Architectures**
>
> We agree that evaluating on advanced transformer-based architectures is important. To address this point, we also evaluated our method on SD3, whose architecture is DiT-based ("Our architecture builds upon the DiT" [6]). The results indicate that our method is effective on DiT-based architectures as well.
>
> &nbsp;
>
> **Q5. Lack of Image Quality Metrics**
>
> We measure image quality using CLIP-I and DINO, which are standard metrics for personalization. We also include perceptual assessments of image quality in the human evaluation (Appendix J, lines 336–367).
>
> &nbsp;
>
> **Q6. Additional Comparison with Previous Works**
>
> As noted in lines 116–117, we also compare our approach with other methods that focus on the relation between the subject and its superclass and follow the prior-preservation loss used in DreamBooth [7, 8]. Our work shows that this scheme does not provide a clear guarantee for maintaining the semantic relation during personalization. Based on this observation, we propose a new objective that directly targets this issue and explores its applicability across standard baselines and different backbone models.
>
> &nbsp;
>
> **Closing Remark**
>
> We appreciate the reviewer's detailed comments and would be happy to engage in further discussion if needed.

---

> > ### Author Response · Authors · 2025-11-19
> > **Response to Reviewer xLt7 (Reference)**
> >
> > ### **Reference**
> >
> > [1] Purushwalkam, Senthil, et al. "Bootpig: Bootstrapping zero-shot personalized image generation capabilities in pretrained diffusion models.” ECCV, 2024.
> >
> > [2] Pan, Xichen, et al. "Kosmos-G: Generating Images in Context with Multimodal Large Language Models." ICLR. 2024.
> >
> > [3] Aiello, Emanuele, et al. "DreamCache: Finetuning-Free Lightweight Personalized Image Generation via Feature Caching." CVPR. 2025.
> >
> > [4] Liu, Mushui, et al. "TFCustom: Customized Image Generation with Time-Aware Frequency Feature Guidance." CVPR. 2025.
> >
> > [5] Ruiz, Nataniel, et al. "Hyperdreambooth: Hypernetworks for fast personalization of text-to-image models." CVPR. 2024.
> >
> > [6] Esser, Patrick, et al. Scaling rectified flow transformers for high-resolution image synthesis. ICML 2024.
> >
> > [7] Huang, Jiannan, et al. "Classdiffusion: More aligned personalization tuning with explicit class guidance." arXiv. 2024.
> >
> > [8] Qiao, Pengchong, et al. "Facechain-sude: Building derived class to inherit category attributes for one-shot subject-driven generation." CVPR. 2024.

---

### Official Review · Reviewer_LdxF · 2025-11-01

**Soundness:** 3
**Presentation:** 3
**Contribution:** 3
**Rating:** 6
**Confidence:** 3

**Summary:**

The paper targets overfitting and distributional drift in few-shot personalization of text-to-image diffusion models. It argues that existing objectives (plain denoising, prior-preservation via class prompts) are misaligned with the dual goals of subject fidelity and text alignment because they provide no explicit guarantee of preserving the pretrained distribution. The core contribution is a Lipschitz-based regularization objective that bounds the deviation of the personalized model from the pretrained one.

**Strengths:**

1 Clear objective-level insight: reframes personalization as a distribution-preserving adaptation problem and exposes misalignment of standard/“prior-preservation” losses with distributional stability.

2 Simple, principled, and practical: the Lipschitz argument leads to a tractable L2 parameter-distance regularizer that is backbone-agnostic, easy to implement, and removes pre-sampling overhead.

3 Solid empirical results: consistent quantitative gains on multiple backbones and personalization strategies; qualitative examples show improved subject identity without sacrificing prompt following. The training-time savings are compelling for practice.

**Weaknesses:**

1 Assumptions and tightness: The Lipschitz continuity of εθ w.r.t. θ and the resulting KL bound are plausible but high-level; constants (λ) and norm choices are not characterized for realistic models. The theory justifies using parameter-distance regularization but provides limited guidance on magnitude selection beyond empirical tuning.

2 Failure analysis: Some failure cases remain (identity structure, prompt compositionality). It would help to characterize when the regularizer helps or hurts (e.g., highly stylized subjects, complex relational prompts).

**Questions:**

1 Human evaluation: Could you expand user studies (more raters/prompts) and include identity verification (face/instance retrieval), and prompt adherence judged by humans, to validate metric conclusions?

---

> ### Author Response · Authors · 2025-11-19
> **Response to Reviewer LdxF**
>
> **Q1. Clarifying Lipschitz Assumptions and Practical Regularization (Assumptions and Tightness)**
>
> The proposed method is theoretically supported, and empirical tuning is expected in practical personalization scenarios. To clarify this relationship between theory and practice, we provide the following explanation.
>
> First, computing the Lipschitz constant of $\theta$ in Theorem 2 requires access to the pretrained data distribution $(x, c)$. However, the personalization task provides only a handful of subject images, making it impossible to recover the characteristics of the pretrained distribution needed to estimate the exact Lipschitz constant of $\theta$.
>
> Second, although $\lambda$ is motivated by the Lipschitz property described in Theorem 2, its practical value in personalization is determined by the characteristics of each subject. As shown in line 9 of Algorithm 1, the regularizer coefficient $\lambda$ balances the new learning signal and the preservation of the pretrained capacity. Since learning dynamics differ across subjects, the optimal $\lambda$ varies as well. We note that choosing a different $\lambda$ only changes the strength of the regularization and does not violate the bound on distribution preservation established in Theorem 2.
>
> Finally, we adopt the $\ell_2$ norm as clarified in Remark 3. We observed that using the raw absolute norm $|\cdot|$ leads to unstable updates, whereas using $|\cdot|_2^2$ consistently yields stable training. This suggests that the $\ell_2$ formulation provides more reliable optimization behavior by assigning larger weight to larger parameter deviations.
>
> &nbsp;
>
> **Q2. Failure Analysis**
>
> We acknowledge the importance of analyzing failure cases. While our method alleviates the common trade-off between image fidelity and prompt alignment, some challenging cases still remain, such as stylized subjects and minor concepts. In Figure 11 of the Appendix L, we observe that our method struggles with cartoon-style minor concepts similar to other approaches. Figure 12 further shows that prompts requiring changes in the subject's appearance are also difficult to handle. These cases indicate that more advanced and adaptive strategies for subject-specific personalization may still be needed, as discussed in Q1 and Appendix A. We will provide additional analysis of these failure cases in the revised version.
>
> &nbsp;
>
> **Q3. Expanded Human Evaluation**
>
> Following reviewer LdxF's suggestion, we expanded the human evaluation in Appendix J. We recruited 15 additional participants for the detailed criteria, resulting in a total of 35 participants for the overall evaluation.
>
> The criteria are as follows:
>
> - **Identity Preservation**: Whether the identity of the user-specified object is well preserved.
> - **Prompt Coherence**: Whether the condition described in the prompt is accurately reflected.
> - **Image Quality**: Whether the image is free of artifacts or distortions.
> - **Overall Preference**: Determined by jointly considering the three criteria above.
>
> The results are shown in Table 1 below. The detailed criteria reveal clear tendencies of each method. DreamBooth (DB) shows strong preference in identity preservation but lower prompt adherence. In contrast, DB-LoRA shows lower identity preservation but twice higher prompt adherence. These tendencies are consistent with the known behaviors of the two methods: DB tends to overfit, whereas DB-LoRA tends to underfit (lines 98–104, 360–364). IP-Adaptor produces good image quality but remains weak in personalization (lines 365–366). Our method maintains a balanced performance across all criteria. This suggests that the proposed objective captures subject-specific information while preserving the pretrained distribution, leading to consistent and stable results. These detailed criteria complement our earlier results and further support the effectiveness of the proposed method.
>
> **Table 1. User study results across four criteria (all values in %).**
>
> |   Method   | Identity Preservation | Prompt Coherence | Image Quality | Overall Preference |
> |:----------:|:---------------------:|:----------------:|:-------------:|:------------------:|
> | **Ours**    | 44.22   | 43.78  | 44.44   |       45.81 |
> | DB          | 33.33   | 23.11  | 24.89   |       25.14 |
> | DB-LoRA     | 11.78   | 22.67  | 13.11   |       17.52 |
> | IP-Adaptor  | 10.67   | 10.44  | 17.56   |       11.53 |
>
> &nbsp;
>
> **Closing Remark**
>
> We appreciate the reviewers' constructive feedback, which has helped improve the clarity and completeness of our work. We will incorporate all relevant suggestions into the final manuscript.

---

### Author Response · Authors · 2025-12-01
**Summary of Reviews and Responses**

This brief summary is provided to clarify the current status of the submission.

**Current Status**
We provided point-by-point responses to the reviewers’ concerns in our rebuttal, and no further replies were received thereafter. We note that the reviewers acknowledged the novelty of our work and requested additional analysis and clarification rather than raising critical concerns. We have addressed each point thoroughly and believe that our responses adequately clarify the reviewers’ questions.

&nbsp;

**Summary of Strengths**
We sincerely appreciate the reviewers’ thoughtful evaluations. We are pleased that they highlighted the clear motivation and principled approach (`LdxF`, `xLt7`, `Bv3p`), the comprehensive experiments (`LdxF`, `Bv3p`), and the well-structured writing of the manuscript (`xLt7`, `Bv3p`) as key strengths.

&nbsp;

**Main Reviewer Concerns & Our Responses**

- **Clarifying the Lipschitz assumptions and their practical role in personalization**
We clarify how the proposed Lipschitz property applies to the personalization. We further discuss its connection to advanced methods, explain why certain approaches are not applicable in our setting, and provide the corresponding implementation details.

- **Providing a detailed user study**
We strengthen the human evaluation by introducing finer-grained criteria and additional participants to more effectively analyze perceptual quality.

- **Improving clarity in writing and analysis**
We refine expressions that may cause ambiguity and describe failure cases more clearly to improve the overall completeness of the manuscript.

In addition, we guided reviewers to the relevant sections of the manuscript to facilitate easier reference to existing details.


&nbsp;

**Key Contributions**
- We provide a theoretical proof that the widely used objective in personalized diffusion models does not guarantee distribution preservation.
- Motivated by this finding, we introduce a personalized objective grounded in the Lipschitz property via parameter regularization, which has not been employed in previous personalization methods.
- Our comprehensive experiments demonstrate that the proposed approach is intuitive and effective compared to existing methods and across diverse backbone models.
- Taken together, this principled objective-level perspective is not tied to a specific model architecture or personalization method, suggesting flexibility in diverse settings and potential extension to other diffusion tasks.

&nbsp;

**Closing Remark**
A revised manuscript reflecting these clarifications is provided. We once again appreciate the reviewers’ constructive feedback, which has helped improve the quality of the manuscript.

---

### Meta-Review · Area_Chair_YwaT · 2026-01-05

**Summary:**

This paper work on personalized text-to-image diffusion model. Authors introduced a Lipschitz-based regularization objective that constrains parameter updates during personalization, ensuring bounded deviation from the original distribution, which promotes consistency with the pretrained model’s behavior while enabling accurate adaptation to new concepts. Experimental results verified the effectiveness of the proposed method.

This paper got two 6 ratings and one 4 rating.

The strength of this paper given by reviewers are:
1. Clear objective-level insight. (Reviewer LdxF)
2. Simple, principled, and practical. (Reviewer LdxF)
3. Solid empirical results. (Reviewer LdxF)
4. paper is well written. (Reviewer xLt7)
5. proposed method is straightforward. (Reviewer xLt7)
6. Clear statement of objective–goal misalignment. (Reviewer Bv3p)
7. Theory sketch linking parameter Lipschitzness → KL control (Theorem 2); connects elegantly to L2 surrogate. (Reviewer Bv3p)
8. Consistent empirical lift on fidelity (DINO/CLIP-I) with competitive CLIP-T across multiple backbones. (Reviewer Bv3p)

The weakness of this paper given by reviewers are:
1. Assumptions and tightness. (Reviewer LdxF)
2. Failure analysis. (Reviewer LdxF)
3. tuning-based personalization (like, DreamBooth, Custom Diffusion) is outdated. (Reviewer xLt7)
4. How did you obtain the results of Fig. 2. (Reviewer xLt7)
5. explain more about the claim "Remark 1.". (Reviewer xLt7)
6. need comparison results based on the DiT-based model. (Reviewer xLt7)
7. evaluation lacks the image quality metrics. (Reviewer xLt7)
8. comparing the proposed methods with previous work. (Reviewer xLt7)
9. why authors' variant is preferable in diffusion personalization should be made explicit. (Reviewer Bv3p)
10. more analysis on prompt adherence failures (per failure figs) would help. (Reviewer Bv3p)
11. Why is this error proportion so large? (Reviewer Bv3p)

Questions:
1. Human evaluation. (Reviewer LdxF)
2. Can you quantify Theorem 2’s $\lambda$ (constant) for a given backbone (e.g., SD-1.5) via empirical Lipschitz estimation, and relate it to your best-performing $\lambda$ in training? (Reviewer LdxF)

AC read authors' paper, reviewers' comments, and authors' rebuttal carefully, found authors addressed reviewers' concerns (details are in the below session), and decide to accept the paper.

**Reviewer Concerns:**

weakness 1. authors gave good justification for this.

weakness 2. authors methods they provided results in Figure 11 and Figure 12. they also willing to provide more results in revised version.

weakness 3. authors provided justification why tuning-based methods still needed.

weakness 4. authors gave details on how they obtain the results of Fig. 2.

weakness 5. authors revised the text to make it less confusing.

weakness 6. authors do use DiT-based model.

weakness 7. authors added image quality in the human evaluation.

weakness 8. authors compared with other methods [7,8]

weakness 9. authors added more explanations.

weakness 10. authors do have some results in the paper and will make the connections more explicitly.

weakness 11. authors mentioned there is a typo and they will fix it.

question 1. authors added additional human evaluation showing their method is better.

question 2. authors added more details.

**Reviewer Scores:**

Reviewer LdxF might keep or increase their score 6.

Reviewer xLt7 might keep or increase their score 4.

Reviewer Bv3p might keep or increase their score 6.

---

### Decision · Program_Chairs · 2026-01-26

Accept (Poster)